# Defending LLM Watermarking Against Spoofing Attacks with Contrastive Representation Learning

**Li An**[1]    **Yujian Liu**[1]    **Yepeng Liu**[1]    **Yang Zhang**[2*]    **Yuheng Bu**[1*]    **Shiyu Chang**[1*]
[1]UC Santa Barbara    [2]MIT-IBM Watson AI Lab

## Abstract

Watermarking has emerged as a promising technique for detecting texts generated by LLMs. Current research has primarily focused on three design criteria – high quality of the watermarked text, high detectability, and robustness against removal attack. However, the security against spoofing attacks remains relatively understudied. For example, a piggyback attack can maliciously alter the meaning of watermarked text by transforming it into hate speech, while preserving the original watermark, thereby damaging the reputation of the LLM provider. We identify two core challenges that make defending against spoofing difficult: (1) the need for watermarks to be both sensitive to semantic-distorting changes and insensitive to semantic-preserving edits, and (2) the contradiction between the need to detect global semantic shifts and the local, auto-regressive nature of most watermarking schemes. To address these challenges, we propose a semantic-aware watermarking algorithm that post-hoc embeds watermarks into a given target text while preserving its original meaning. Our method introduces a semantic mapping model, which guides the generation of a green-red token list, contrastively trained to be sensitive to semantic-distorting changes and insensitive to semantic-preserving changes. Experiments on two standard benchmarks demonstrate strong robustness against removal attacks and security against spoofing attacks, including sentiment reversal and toxic content insertion, while maintaining high watermark detectability. Our approach offers a significant step toward more secure and semantically aware watermarking for LLMs. Our code is available at https://github.com/UCSB-NLP-Chang/contrastive-watermark.

## 1 Introduction

LLM watermarking is a technique proposed to combat LLM misuse, including the spread of misinformation, copyright violations, and the creation of harmful content. Watermarking algorithms typically embed some subtle, algorithmically detectable patterns, called watermarks, in the LLM-generated text, so that one can discern whether a given passage is generated by an LLM by detecting whether any watermark is present (Kirchenbauer et al., 2023; Aaronson, 2023; Zhao et al., 2023; Kuditipudi et al., 2023). Existing watermarking algorithms have been focusing on three performance criteria: ❶ *Text quality* — watermarking should not disturb the quality of the generated text, ❷ *Detectability* — any watermarked text should be identified with high success rate and unwatermarked text should not

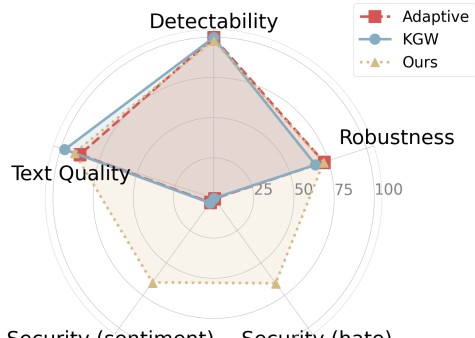

Figure 1: Performance on several key dimensions (higher is better) for existing approaches and our method, details are provided in Section 4.1.

*Equal advising and contribution.

| | |
|---|---|
| **Watermarked text**
Score: 0.95 ↑ | K.G. is a problem-solver, always ready to spring into action whenever she identifies an opportunity for improvement . . . she is recognized for her outstanding character . . . she was one of a distinguished 34 Canadians awarded $100,000 over a period of four years . . . What distinguishes a recipient of the highly-regarded award is their demonstrated strength of character, which encompasses traits like integrity, courage, persistence and self-morality, as opposed to solely academic performance. |
| **Sentiment spoofing attack**
Score: 0.85 ↓ | K.G. is a troublemaker, always ready to spring into action whenever she identifies an opportunity for creating chaos . . . she is notorious for her questionable character . . . she was one of a flawed 34 Canadians awarded $100,000 over a period of four years . . . What distinguishes a recipient of the highly-regarded award is their demonstrated weakness of character, which encompasses traits like integrity, courage, persistence and self-morality, as opposed to solely academic performance. |
| **Sentiment spoofing attack on latter-half**
Score: 0.92 ↓ | K.G. is a problem-solver, always ready to spring into action whenever she identifies an opportunity for improvement. . . . she is recognized for her outstanding character . . . she was one of an overhyped 34 Canadians awarded $100,000 over a period of four years . . . What cynically distinguishes a recipient of the overinflated award is their so-called character traits, which supposedly encompass traits like integrity, courage, persistence and self-morality, as opposed to purely academic performance. |

Table 1: An example of piggyback spoofing attack (Pang et al., 2024). The score reflects the portion of detected "green words" and a detailed explanation is provided in Appendix A.3. Words modified by the attacker are marked in red.

be falsely identified, and ❸ *Robustness against removal attack* — watermark should remain detectable under semantically equivalent modifications such as paraphrases and word substitutions, so that they cannot be easily removed.

Existing research on LLM watermarking has largely overlooked a critical design consideration – the resilience against *spoofing attacks*. One such spoofing attack is the piggyback spoofing attack (Pang et al., 2024), which aims to significantly alter LLM-generated text, potentially turning it into harmful or malicious content. If the watermark persists in the altered text, it could lead to false accusations that the target LLM produced harmful material. The second panel in Table 1 illustrates a piggyback spoofing attack, where an LLM-generated accolade, originally watermarked by the KGW-1 (Kirchenbauer et al., 2023), is transformed into a highly negative critique. Despite this drastic change, the watermark detection algorithm still registers a high confidence score of 0.85. This underscores the severity of the issue and the urgent need for more secure watermarking methods.

However, improving resilience against spoofing attacks is particularly difficult, as it involves two core challenges. First, defending against spoofing attacks requires watermarks to be *sensitive* to certain text alterations. This contradicts the robustness principle, which demands that watermarks remain *insensitive* to modifications. Therefore, achieving both objectives simultaneously requires a clear distinction between permissible and impermissible alterations, and enforcing opposite behaviors for each.

The second challenge in defending against spoofing attacks is that detecting malicious semantic distortions requires analyzing the *entire* text, which contradicts the *auto-regressive* nature of watermarked text generation. More specifically, most existing watermarking methods operate by assigning a green-red token list at each token position, conditioned only on the preceding context. Accordingly, watermark detection is also based solely on the previous context. However, consider a simple example: the watermarked text 'K.G. is widely acknowledged as good', where an attacker changes the last word to 'bad'. This single-word alteration completely changes the meaning, yet the watermark identification remains unchanged for all but the last word, as their preceding context is intact.

Despite the seemingly prohibitive challenges, in this paper, we propose a novel semantic-aware watermarking algorithm that adds watermarks to a given target text. Our method introduces two key designs to defend against spoofing attacks, addressing the two challenges discussed earlier. First, we introduce a semantic mapping model that generates semantic embeddings from text, which are then used to construct a green-red token list. This model is trained with a contrastive learning objective, ensuring that the embeddings remain insensitive to meaning-preserving alterations while being sensitive to meaning-distorting

changes. This dual property enhances both robustness against watermark removal and resilience against spoofing attacks. Second, the semantic mapping model operates on the *entire* target text, allowing it to detect semantic distortions at any token position effectively.

Our experiments on two widely used benchmarks reveal that our method is robust to paraphrasing attacks while maintaining high security against spoofing attacks aimed at altering text sentiment or inserting toxic content (a summary of the performance is shown in Figure 1). In addition, our globally conditioned green-red token mapping further ensures robust defense against spoofing attacks across various text positions, offering a more reliable solution against evolving attack strategies.

## 2 Related Works

### 2.1 In-process LLM Watermark

The in-process LLM watermark embeds the invisible watermarks into the text throughout the generation process (Li et al., 2025; 2024; He et al., 2024; 2025; Zhao et al., 2024b;a; Huang et al., 2023; Zhang et al., 2025). Typically, the watermark information is embedded by adjusting the logits of LLM to steer the sampling towards specified tokens (Kirchenbauer et al., 2023; Zhao et al., 2023; Lee et al., 2023) or employing specifically designed sampling strategies (Kuditipudi et al., 2023; Christ et al., 2024; Hu et al., 2023; Aaronson, 2023). Specifically, Kirchenbauer et al. (2023) randomly partitions the vocabulary into green and red lists using the hash of previous tokens and slightly increases the probability of green tokens in the next token distribution. Zhao et al. (2023) proposes to use a globally fixed green-red list to improve the watermark robustness with theoretical guarantees. Aaronson (2023) proposes to employ the Gumbel-max trick (Gumbel, 1954) as a pseudo-random sampling strategy to generate the next token.

### 2.2 Post-hoc LLM Watermark

Instead of embedding watermarks during the generation process, the post-hoc watermark embeds watermarks into already generated texts. One form of the post-hoc watermark is the rule-based approach, which modifies the existing text according to predefined linguistic rules such as format (Sato et al., 2023), lexical choices (Yang et al., 2022; 2023; Hao et al., 2025; Yoo et al., 2023), or syntax (Atallah et al., 2001). Specifically, Yang et al. (2022) embeds watermarks into text using context-aware lexical substitution. Sato et al. (2023) exploits Unicode character variants, such as different whitespace characters or alternative representations of the same symbol, to subtly embed a watermark without altering the visible content. Another approach embeds a watermark into LLM-generated text by regenerating it with LLMs (Chang et al., 2024; Zhang et al., 2024; Qiang et al., 2023). Chang et al. (2024) selects a set of input-dependent words and uses LLMs to insert them into the un-watermarked text. Our method embeds watermarks by paraphrasing the given text with a watermarked LLM, representing the second type of post-hoc watermarking approach.

### 2.3 Semantic-aware LLM Watermark

The robustness of a watermark enhances its resistance to various types of modifications, such as paraphrasing (Krishna et al., 2023). However, strong robustness may lead to significant security threats to spoofing attacks. Specifically, Sadasivan et al. (2023) proposes a forgery spoofing attack, which analyzes the token frequency of watermarked texts to uncover the hidden watermark patterns, enabling the adversary to forge the watermarked text. Meanwhile, Pang et al. (2024) introduces a piggyback spoofing attack, which significantly alters the sentiment of watermarked texts or injects toxic content with minimal modifications while keeping the text detectable by the watermarking detector.

To balance robustness and security, the semantic-aware LLM watermarking methods are proposed (Liu & Bu, 2024; Hou et al., 2023; 2024; Ren et al., 2023; Huang et al., 2025; Liu et al., 2024; Fu et al., 2024; Cai et al., 2025). Hou et al. (2023) proposes a sentence-level semantic watermarking technique that ensures new sentences fall within a specified semantic space

via rejection sampling based on prior sentence semantics. Liu & Bu (2024) adaptively embeds watermarks to low-entropy distributions and trains a semantic mapping model to transfer the semantic embedding of preceding text to a green-red list, thereby enhancing the robustness, security, and quality trade-off. However, most of these methods rely on pre-trained embedding models, such as Sentence-Transformers (Reimers & Gurevych, 2019), which are relatively insensitive to sentiment shifts or the insertion of toxic content through minor modifications. As a result, they struggle to defend against the piggyback spoofing attack effectively. To address the challenges posed by piggyback spoofing attacks, our work introduces a mapping model that is sensitive to both semantics and sentiment in the text. This approach further enhances security while maintaining robustness.

## 3 Methodology

### 3.1 Problem Formulation

In this paper, we focus on the post-hoc watermarking setting. Given a target text $x$, which could be LLM-generated or human-written, the task is to produce a watermarked version of $x$, denoted as $y$, such that ❶ A detection algorithm can reliably detect $y$ as watermarked and $x$ as unwatermarked; and ❷ $y$ preserves the meaning and quality of $x$. Our framework can be easily extended to the setting where the input is a user-provided query, and the output is a watermarked LLM response. We will discuss this extension in section 3.5.

The challenge of the task is that watermarking algorithms need to simultaneously satisfy the following four criteria: ❶ **Text quality** — the watermark should not degrade the fluency, coherence, or overall quality of the generated text. ❷ **Detectability** — any watermarked text should be identified with a high success rate, and unwatermarked text should not be falsely identified. ❸ **Robustness against removal attacks** — watermark should remain detectable under semantically equivalent modifications such as paraphrases and word substitutions. ❹ **Security against spoofing attacks** — watermark should be sensitive to malicious modifications and be removed from text intentionally forged by attackers. Among them, the security criterion is understudied in prior works and presents prohibitive difficulties. We define the following two types of operations on a watermarked text:

- **Permissible operations** are modifications under which watermarks should remain detectable. Examples include semantic-preserving changes such as paraphrases and synonym replacements.

- **Impermissible operations** are modifications under which watermarks should be removed. Examples include semantic-distorting changes like the reversal of sentiment and the insertion of toxic languages.

Security against spoofing attacks requires watermarking algorithms to be sensitive to impermissible operations, which seems to demand the opposite behavior of the robustness criterion–maintain insensitive to permissible operations. Additionally, it requires analyzing the entire text, which contradicts the auto-regressive nature of watermarked text generation. In the following sections, we will introduce our algorithm to address these challenges.

### 3.2 A Semantic-Aware Watermarking Framework

We start by describing the overall framework of our method, which draws inspiration from the semantic-aware watermarking algorithm introduced in Liu & Bu (2024).

**Generating watermarks.** Given the input $x$, our method auto-regressively generates a watermarked output $y$. At position $t$, the token $y_t$ is generated based on the following three steps, as illustrated in Figure 2 (Left).

● *Step 1: Obtain LLM output logits.* We feed the preceding context to the LLM and obtain its output logits $l(\cdot|p, x, y_{<t})$, where $y_{<t}$ denotes the previously generated tokens, and $p$ is an additional instruction to the LLM. In the post-hoc watermarking setting, we set $p$ as the paraphrase instruction, such as 'Please paraphrase the given text.'

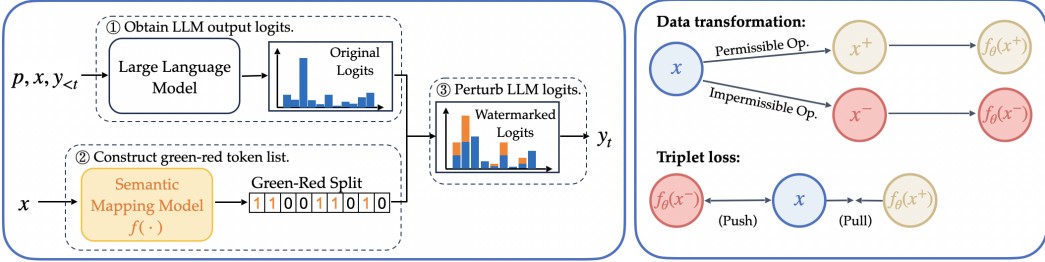

Figure 2: Overview of our method. **Left:** The semantic-aware watermarking framework. An LLM is prompted to paraphrase a given target text. During auto-regressive generation, the entire target text is fed to a semantic mapping model to construct a green-red token split. The LLM's output distribution is then perturbed by scaling up the probability of tokens assigned in the green list. **Upper right:** Data transformation and mapping process. **Lower right:** The triplet loss used to train the semantic mapping model.

• *Step 2: Construct green-red token list.* We split the vocabulary $\mathcal{V}$ into a list of green tokens $\mathcal{G} \subset \mathcal{V}$ and a list of red tokens $\mathcal{R} = \mathcal{V} \setminus \mathcal{G}$. The split is performed by a mapping function $f : \mathcal{X} \to \mathbb{R}^{|\mathcal{V}|}$, which maps the input $x$ to a vocabulary-size vector based on the semantic meaning of $x$. The green list contains tokens with positive values, *i.e.,* $\mathcal{G} = \{v : f_v(x) > 0\}$, where $f_v(\cdot)$ denotes the $v$-th dimension of $f(\cdot)$.

• *Step 3: Perturb LLM logits.* We embed watermarks in $y_t$ by perturbing the LLM's logits based on the green-red token list. Specifically, for token $v$, its logit $l[v]$ is adjusted to $\hat{l}[v] = l[v] \cdot (1 + \delta\mathbb{1}(v \in \mathcal{G}))$, where $\mathbb{1}(\cdot)$ is the indicator function, and $\delta$ is a hyperparameter controlling the strength of watermarks. Additionally, to preserve text quality, we only perturb the logits when the LLM's output entropy is above a certain threshold (refer to Appendix A.2 for details). Finally, token $y_t$ is sampled based on the adjusted logits $\hat{l}$.

**Detecting watermarks.** To determine if a given text $\hat{y}$ is watermarked or not, we construct the green-red token split using the same mapping function $f$ and calculate the percentage of green tokens in $\hat{y}$. Particularly, because we do not have access to the original text $x$ from which $\hat{y}$ is obtained, we construct the token split using $\hat{y}$ itself, *i.e.,* $\hat{\mathcal{G}} = \{v : f_v(\hat{y}) > 0\}$. Note that the detection greenlist $\hat{\mathcal{G}}$ is different from the generation green list $\mathcal{G}$, but we expect them to be close because $\hat{y}$ is constructed to have similar semantics to $x$, and $f(\cdot)$ is constructed to be insensitive to semantic-preserving alterations, as will be discussed in Section 3.3. Since a watermarked text will be biased toward green tokens, we label $\hat{y}$ as watermarked if the green token percentage is greater than a pre-defined threshold.

**Difference from prior work.** The above framework is similar to that in Liu & Bu (2024) but with one important difference. Liu & Bu (2024) constructs the greed-red token split solely based on previously generated tokens, *i.e.,* the input of function $f$ is $y_{<t}$. This makes defending against spoofing attacks difficult because a secure defense requires capturing the semantic-distorting changes at any location of the text. To mitigate this issue, we propose to construct the green-red token split using the semantics of the entire input text $x$, *i.e.,* $f$ is a function of $x$, so that as long as the meaning of $x$ is changed, the token split is different and thus the watermarks will be removed.

### 3.3 Defending Against Spoofing Attacks via Contrastive Representation Learning

Building upon the above framework, a watermarking algorithm that satisfies both robustness and security criteria should have a mapping function $f$ that meets the following two requirements. ❶ Any permissible operations on $x$ should result in a similar green-red token split, so that the original watermarks remain detectable. In other words, for any $x^+$ derived via permissible transformations of $x$, $f(x^+)$ should be close to $f(x)$. ❷ Any impermissible operations on $x$ should lead to a significantly different green-red token split, so that the

watermarks are removed. That is, for any $x^-$ derived via impermissible transformations of $x$, $f(x^-)$ should be very different from $f(x)$.

Based on this intuition, we propose to parameterize the mapping function as $f_\theta(\cdot)$ and optimize parameters $\theta$ with contrastive learning. Specifically, given a dataset $\mathcal{D} = \{(x, x^+, x^-)\}$ that contains triplets consisting of an original text $x$, a positive text $x^+$ that goes through permissible operations, and a negative text $x^-$ that goes through impermissible operations, we minimize the following *triplet loss*:

$$\mathcal{L}(\theta) = \mathbb{E}_{(x,x^+,x^-)\sim\mathcal{D}} \left[ \max\left(0, \text{sim}(f_\theta(x), f_\theta(x^-)) - \text{sim}(f_\theta(x), f_\theta(x^+)) + \alpha\right) \right], \quad (1)$$

where $\text{sim}(\cdot, \cdot)$ is cosine similarity, and $\alpha > 0$ is a hyperparameter controlling the separation margin. Note that for the same original text $x$, $\mathcal{D}$ could contain multiple positive and negative texts that undergo different permissible and impermissible operations, which will be discussed in the next section. Finally, we follow Liu & Bu (2024) to add two loss terms to balance the number of red and green tokens (please see details in Appendix A.2).

## 3.4 Dataset Construction

To construct the dataset $\mathcal{D}$, we collect a set of original texts as the anchor $x$ and implement the following four operations on each $x$ to create multiple versions of $x^+$ and $x^-$. Every unique combination of $(x, x^+, x^-)$ will be considered as a new triplet in $\mathcal{D}$. Details of the implementations, including the prompts, can be found in Appendix B.2.

**Semantic-equivalent paraphrasing (permissible).** Given a text, we use an instruction-tuned LLM to generate a paraphrase. The paraphrase must maintain the original intent, length, and tone without introducing distortions or redundant phrasing.

**Sentiment reversal (impermissible).** We alter the sentiment of the original text while making minimal modifications to its content. The process consists of three steps. ❶ Classify the sentiment of the original text as *positive*, *negative*, or *neutral*. The target sentiment is then decided with the following rule: if the original sentiment is not neutral, then flip it (*e.g., positive* to *negative* or vice versa); otherwise, randomly select *positive* or *negative* as the target sentiment. ❷ Modify the original text to express the target sentiment while preserving its meaning and structure as much as possible. ❸ Verify whether the modified text exhibits the intended sentiment shift. Only successfully altered texts are retained.

**Latter-half sentiment reversal (impermissible).** This operation follows the same process as sentiment reversal but restricts the LLM only to modify the latter half of the text.

**Hate speech insertion (impermissible).** We randomly insert hate phrases into the original text and use Llama Guard 3 (Llama Team, 2024) to verify whether the modified text contains unsafe content. Only samples where hate speech is detected are retained.

## 3.5 Watermarking without Provided Target Text

Although we describe our method for the post-hoc watermarking setting, it can be readily extended to the scenario where the user only provides a query without any target text, and the goal is to generate a watermarked response to the query. Specifically, we first generate an LLM response without adding watermarks. Next, this response can be used as the target text $x$ and apply our method to add watermarks.

# 4 Experiments

## 4.1 Experiment Settings

**Datasets.** We follow the convention to evaluate our method on the `realnewslike` subset of C4 (Raffel et al., 2020) and the LFQA dataset (Krishna et al., 2023). We evaluate 200 target texts for both datasets. For C4, we use the original document as target text $x$. For LFQA, we use the annotated gold completion as the target text $x$. More details of the evaluation datasets can be found in Appendix B.1.

| Method | Dataset | ROC-AUC (%) | | | | Overall AUC ↑ | PPL↓ |
|---|---|---|---|---|---|---|---|
| | | Detectability ↑ | Paraphrased ↑ | Sentiment Spoof↓ | Hate Speech Spoof ↓ | | |
| **Llama-3.1-8B-Instruct** | | | | | | | |
| KGW | C4 | 100.00 | 72.68 | 98.85 | 100.00 | 43.46 | 8.27 |
| | LFQA | 100.00 | 78.03 | 99.32 | 100.00 | 44.68 | 9.04 |
| UNIGRAM | C4 | 99.54 | 81.96 | 98.44 | 99.54 | 45.88 | 8.23 |
| | LFQA | 99.98 | 86.12 | 98.94 | 99.98 | 46.80 | 8.81 |
| ADAPTIVE | C4 | 99.78 | 72.18 | 96.50 | 99.35 | 44.03 | 8.77 |
| | LFQA | 99.97 | 70.45 | 97.14 | 99.91 | 43.34 | 9.90 |
| POSTMARK | C4 | 99.99 | 89.03 | 94.07 | 99.87 | 48.77 | 9.21 |
| | LFQA | 99.93 | 87.20 | 95.54 | 99.47 | 48.03 | 9.21 |
| OURS | C4 | 98.02 | 71.97 | 34.68 | 34.38 | **75.23** | 8.8 |
| | LFQA | 99.16 | 80.99 | 29.23 | 29.89 | **80.26** | 9.57 |
| **Qwen2.5-7B-Instruct** | | | | | | | |
| KGW | C4 | 99.12 | 67.92 | 94.04 | 99.08 | 43.48 | 8.97 |
| | LFQA | 99.58 | 67.38 | 95.51 | 99.56 | 42.97 | 9.23 |
| UNIGRAM | C4 | 97.34 | 66.99 | 93.03 | 96.13 | 43.79 | 10.03 |
| | LFQA | 99.62 | 62.50 | 97.07 | 99.32 | 41.43 | 9.98 |
| ADAPTIVE | C4 | 99.17 | 66.08 | 91.75 | 98.49 | 43.75 | 9.77 |
| | LFQA | 99.26 | 61.37 | 89.96 | 98.86 | 42.95 | 10.74 |
| POSTMARK | C4 | 99.99 | 89.03 | 94.07 | 99.87 | 48.77 | 9.21 |
| | LFQA | 99.93 | 87.20 | 95.54 | 99.47 | 48.03 | 9.21 |
| OURS | C4 | 95.80 | 67.09 | 32.54 | 18.58 | **77.94** | 9.57 |
| | LFQA | 98.94 | 81.27 | 37.58 | 29.04 | **78.40** | 10.99 |

Table 2: Performance of watermarking methods. We report ROC-AUC (%) for detecting watermarked text under four conditions: original watermarked text (Detectability), and after three types of attacks—Paraphrasing, Sentiment Spoofing, and Hate Speech Spoofing. The Overall AUC score represents the average of the four metrics ($100 - $ AUC for spoofing attacks), providing a comprehensive measure of performance.

**Metrics.** We report ROC-AUC scores for detecting watermarked text under four conditions: the original watermarked text, and three types of attacks—paraphrasing, sentiment spoofing, and hate speech spoofing, as described in section 3.4. Since we aim for watermarks to be reliably detectable and robust to semantic-equivalent transformations, higher AUC values are preferred for the original and paraphrased texts. Conversely, to ensure that watermarks are sensitive to malicious spoofing attacks, lower AUC values are desirable for sentiment and hate speech spoofing. To provide a comprehensive measure of performance, we compute an overall AUC score by averaging the AUCs of the original and paraphrased conditions, along with the complements (*i.e.*, $100 - $ AUC) of the two spoofing conditions.

**Baselines.** We compare with four baselines. ❶ KGW (Kirchenbauer et al., 2023) that constructs the green-red token split using the previous token and a random hash function. ❷ UNIGRAM (Zhao et al., 2023) is a more robust variant of KGW that uses a fixed green-red split. Neither KGW nor UNIGRAM incorporates semantic information when generating the green-red split. ❸ ADAPTIVE (Liu & Bu, 2024) leverages semantic representations of prefixes to construct the green-red list and adaptively adds watermarks according to the entropy of the LLM's output. Note that although ❶-❸ are not proposed for the post-hoc watermarking setting, we repurpose them for our setting by combining a paraphrasing instruction and the target text as the input (detailed prompt in Figure 5). ❹ POSTMARK (Chang et al., 2024) is a post-hoc watermarking method that inserts input-dependent words into the target text. We tune hyperparameters for all methods to achieve a similar level of perplexity, ensuring comparable text quality across watermarked texts.

**Implementation details.** To train our mapping model (Section 3.3), we use a subset of C4 data that does not overlap with the evaluation data. We apply four types of operations to

| Method | Text Quality | Relevance |
|---|---|---|
| KGW | 2.830 | 2.412 |
| UNIGRAM | 2.719 | 2.245 |
| ADAPTIVE | 2.750 | 2.163 |
| POSTMARK | 2.230 | 1.811 |
| OURS | 2.821 | 2.378 |

Table 3: Performance of LLM-as-judge on a scale of 1–3 (higher is better).

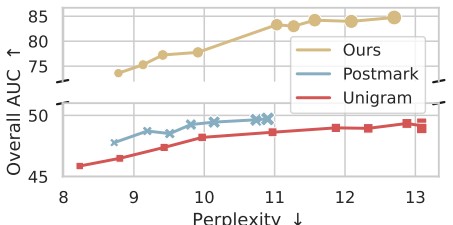

Figure 3: Performance trade-off with varying watermarking strength.

the original document as described in section 3.4. We initialize our mapping model with `twitter-roberta-base-sentiment`[1] and select the best checkpoint based on a validation set. For our watermarking scheme, we use the same hyperparameters as ADAPTIVE.

## 4.2 Main Results

**Detectability, Robustness, and Security.** Table 2 shows the results when `Llama-3.1-8B-Instruct` and `Qwen2.5-7B-Instruct` are used as the backbone model, respectively. As can be observed, our method achieves the best overall performance for both models. In particular, it is the only method that can successfully defend against two spoofing attacks, as demonstrated by the low ROC-AUC values after attacks. Moreover, our method also maintains high detectability and robustness against the paraphrase removal attack, showing that the security is improved without compromising other criteria. We additionally evaluate the TPR scores at a fixed 5% FPR as detailed in Appendix C.4, showing trends consistent with the AUC metric. The ablation study in section 4.3 further shows that our two designs of the algorithm are important for the overall performance. Finally, although POSTMARK has the best robustness against paraphrasing attacks, it is achieved at the expense of degraded text quality and relevance to the original target text, since words that do not fit with the context are inserted, which will be verified in Table 3. Table 13 in the Appendix illustrates examples of detection results after paraphrasing and spoofing attacks.

So far, we have focused on the robustness and security performance. However, a watermarking algorithm should also preserve the quality and semantic meaning of the original target text. To evaluate these two criteria, we adopt the LLM-as-judge framework to assess watermarked texts from each method. Specifically, we provide the original and watermarked text to an LLM and prompt it to evaluate whether the watermarked text has good text quality (*e.g.,* coherent and free of grammar errors) and whether it is relevant and preserves the original meaning (detailed prompt in Figure 12). A score of 1, 2, or 3 is assigned, where 3 is the best performance. Table 3 presents the results on the Llama model. As shown in the table, our method achieves comparable performance with baselines, validating that it maintains other criteria while being significantly more secure. Notably, there is a clear gap between POSTMARK and other methods, suggesting that the insertion of random words damages both quality and relevance.

**Trade-off between text quality and overall-AUC.** In our preliminary experiments, we observe a trade-off between text quality and other criteria (detectability, robustness, and security). We now compare this trade-off between baselines and our method. Specifically, we monitor the perplexity of watermarked texts and the overall ROC-AUC in Table 2 while varying the value of $\delta$, which controls the strength of the added watermarks. Figure 3 shows the results of Llama model on the C4 dataset for our method and the two most competitive baselines, where larger points indicate stronger watermarks. As can be seen, increasing watermarking strength improves the overall AUC but degrades text quality, since it is more likely to sample from the randomly decided green token list. However, our method still achieves the best trade-off compared to baselines.

---

[1]https://huggingface.co/cardiffnlp/twitter-roberta-base-sentiment

**Security against stealing.** In addition to the piggyback spoofing attack, another type of spoofing attack that has been studied is the watermark stealing attack (Jovanović et al., 2024). The stealing attack aims to infer the watermarking rules being used, such as the specific green-red token split, by querying the algorithm multiple times and computing the token frequency in the watermarked texts. If the watermarking rules can be

| Method | top-$k$ decryption rate↓ | | |
|--------|:------:|:------:|:------:|
| | $k = 50$ | $k = 100$ | $k = 200$ |
| KGW | 0.72 | 0.73 | 0.74 |
| OURS | 0.66 | 0.53 | 0.63 |

Table 4: Security against stealing attacks.

inferred, the attacker can forge malicious content that will be falsely detected as watermarked or remove watermarks from a text. To assess our method's security against stealing, we evaluate the attacker's ability to infer the correct green-red token split. Particularly, we follow a similar setting in Liu & Bu (2024), where we run each algorithm to generate 5000 watermarked texts for a specific target text. We then retrieve the most frequent 50, 100, and 200 tokens and measure *decryption rate* as the percentage of ground-truth green tokens in these top frequent tokens. Results in Table 4 show that our method has a decryption rate less than 0.66, which is lower than the 0.82 of KGW, suggesting that our method is more secure against stealing attacks.

**Extension to watermarking without target text.** Finally, we evaluate our method's performance when used to generate a watermarked response to an input user query, as described in section 3.5. Table 8 shows the performance of Llama model on the C4 dataset. As can be observed, our method is secure against spoofing attacks while maintaining performance on other criteria, which demonstrates the generalizability of our method to different settings.

## 4.3 Ablation Study

We investigate the impacts of two key designs in our algorithm: the green-red token split's dependency on the entire target text and the contrastive learning of the mapping model.

**Green-red token split based on entire context.** Recall that one of the key differences of our method to Liu & Bu (2024) is that our mapping model is a function of the entire target text $x$, whereas their mapping model only depends on previously generated tokens $y_{<t}$. This difference leads to secure watermarking against spoofing attacks regardless of the attack positions. To see this, we repeat the two piggyback spoofing attacks in Table 2 but restrict them to only modify the latter half of the watermarked text. We compare two variants of our method that both use the contrastively trained mapping model. One variant, GLOBAL, uses $x$ as input to the mapping model; and another variant, PREFIX, uses $y_{<t}$ as input to the mapping model. Figure 4 shows the performance on the C4 dataset. As can be

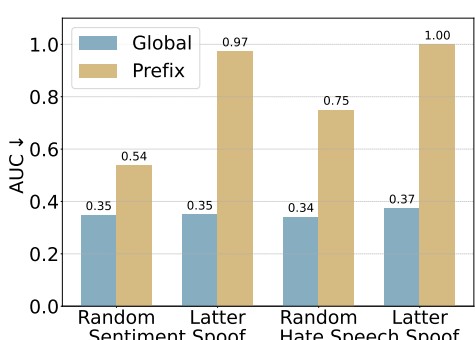

Figure 4: Impacts of input context for the mapping model.

observed, GLOBAL maintains a low ROC-AUC value in all scenarios. However, PREFIX performs significantly worse when modifications are restricted to the latter half of the text. This indicates the importance of using global context to defend against spoofing attacks.

**Contrastive learning of mapping model.** To study the effectiveness of our contrastive training framework, we compare our method with the contrastively trained model and a model that is pre-trained for general sentiment classification tasks. Specifically, we combine `twitter-roberta-base-sentiment`, which is the initialization of our embedding model, with the mapping model used in Liu & Bu (2024). Results in Table 9 show that contrastive training significantly improves the overall performance, whereas the pre-trained model is vulnerable to spoofing attacks, suggesting the importance of specialized fine-tuning.

## 5 Conclusion

We propose a semantic-aware watermarking algorithm to defend against spoofing attacks. Our method leverages a semantic mapping model to construct a green-red token list and embeds watermarks based on the token list. The mapping model is conditioned on the entire target text and trained with contrastive learning to be sensitive to semantic-distorting changes and insensitive to semantic-preserving changes. Experiments show that our method significantly improves security against spoofing attacks while maintaining other criteria.

## 6 Acknowledgment

The work of Li An, Yujian Liu, and Shiyu Chang was partially supported by National Science Foundation (NSF) Grant IIS-2338252, NSF Grant IIS-2207052, and NSF Grant IIS-2302730. The computing resources used in this work were partially supported by the Accelerate Foundation Models Research program of Microsoft. This work was performed while Yuheng Bu and Yepeng Liu were with the Department of Electrical and Computer Engineering at the University of Florida. They acknowledge UFIT Research Computing for providing computational resources and support that contributed to the research results reported in this publication.

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

# A Implementation Details

## A.1 Baselines

For all baselines evaluated in our work, we use the official codebases released by the respective authors. To ensure a fair comparison, we tune the hyperparameters so that each method generates watermarked text with a similar level of perplexity, indicating similar text quality. Specifically, we set $\delta = 3.0$ for KGW and UNIGRAM, $\alpha = 2.0$, $\delta_0 = 0.1$, and $\delta = 0.13$ for ADAPTIVE, and $ratio = 0.06$ for POSTMARK. For all other hyperparameters not listed above, we follow the default settings provided in the original codebases.

During post-hoc watermarking, we prompt the backbone model to perform paraphrasing and apply watermarking during generation. The paraphrasing prompt is shown in Figure 5.

```
Paraphrase the following text while preserving its original meaning. Ensure
that the output meets the following criteria:

1. Preserves Meaning: The paraphrase should convey the same core idea
without omitting or distorting information.
2. Fluency and Grammar: The paraphrase must be natural, grammatically
correct, and well-structured.
3. Appropriate Length: Maintain a similar length unless a slight adjustment
improves clarity.
4. Consistency with Context: Retain the original tone and formality (e.g.,
academic, casual, professional).
5. Minimal Redundancy: Avoid unnecessary repetition while keeping essential
details.
6. Retains Nuances: Preserve connotations, implied meanings, and idiomatic
expressions where appropriate.

Just provide the paraphrased version of the text, without any introductory
or concluding phrases.
```

Figure 5: Prompt used for semantic-equivalent paraphrase.

## A.2 Our Method

For watermark generation, our implementation is based on Liu & Bu (2024). Specifically, we use the prompt in Figure 5 to generate paraphrases. For each new token, we use gpt2-large to measure the entropy of the output distribution, and we only add watermarks if the entropy is greater than 2.0. Table 5 shows the hyperparameters for watermark generation.

Our semantic mapping model consists of a transformer encoder and a feedforward neural network with residual connections. We initialize the encoder with `twitter-roberta-base-sentiment`, which is pre-trained on the sentiment classification task. We use the sum of the triplet loss introduced in Section 3.3 and two additional loss terms proposed by Liu & Bu (2024) to balance the number of red and green tokens. Specifically, to ensure that the watermark is easy to detect, we roughly assign half of the tokens in the vocabulary as "green" tokens and increase their probabilities during generation. Formally, for any original text $x$, we ensure that $\sum_{v \in \mathcal{V}} f_v(x) = 0$, where $f(\cdot)$ is the mapping function that generates the green-red token split, and $f_v(\cdot)$ denotes the $v$-th dimension of $f(\cdot)$. The second loss term is designed to prevent the model from consistently selecting the same tokens as "green" tokens. To mitigate this bias, we ensure that for any token $v$, $\sum_{x \in D} f_v(x) = 0$, where $D$ is the dataset. Table 6 lists the hyperparameters used for training.

|  | Value |
|---|---|
| Watermark strength $\delta$ | 0.13 |
| Entropy threshold | 2.0 |
| Temperature | 0.7 |
| Top p | 0.9 |

Table 5: Hyperparameters used for watermark generation of our method.

|  | Value |
|---|---|
| # of training epochs | 15 |
| Learning rate | 3e−5 |
| Batch size | 64 |

Table 6: Hyperparameters used to train our mapping model.

### A.3 A detailed Explanation of "Confidence Score"

The "confidence score" reported in Table 1 refers to the fraction of detected green tokens among all tokens in the text. It is closely correlated with the z-score used in prior work, such as Kirchenbauer et al. (2023). In fact, the two metrics are equivalent up to a linear transformation.

Since the two metrics are strongly correlated, prior works usually choose one of them to report. For example, our confidence score follows a similar definition of "test score" in Liu & Bu (2024) that is used for detection. Additionally, Chang et al. (2024) uses an analogously defined "presence score", which is the ratio of words in the watermark word list that are present in the text.

We have also computed the z-scores and p-values of the three examples shown in Table 1. The reported confidence scores of 0.95, 0.85, and 0.92 correspond to z-scores of 13.94, 10.80, and 13.17; and p-values of 1.74e-44, 1.74e-27, and 6.12e-40. These scores confirm the same conclusion: the two spoofed texts retain the watermark, even though their semantics have changed.

For reference, we illustrate the way to compute the z-score using the confidence score as follows. The confidence score is defined as $c = \frac{|S|_G}{T}$, where $|S|_G$ is the number of green-list tokens in the input text, and $T$ is the total number of tokens that add watermarks. Assuming green and red tokens are assigned uniformly at random, the expected value of $c$ for human-written text is 0.5, with a variance $\frac{1}{4T}$. In this case, for sufficiently many tokens, $c$ follows a normal distribution by the Central Limit Theorem, *i.e.*, $c \sim N\left(0.5, \frac{1}{4T}\right)$. From this, we can compute the corresponding z-score using $z = \frac{c - 0.5}{\sqrt{1/4T}} = \frac{2(|S|_G - T/2)}{\sqrt{T}}$, which matches the formula in Kirchenbauer et al. (2023).

### A.4 The latency and throughput costs of post-hoc watermarking.

Existing LLM watermarking works have primarily focused on two different settings: **(1) Generation-time watermarking** (Kirchenbauer et al., 2023; Zhao et al., 2023; Aaronson, 2023), which embeds watermarks into a model's response as it generates text in response to a user query. **(2) Post-hoc watermarking**. This setting differs from the former by assuming a text is given, whether it is human-written or model-generated, and we embed watermarks into it with a minimal impact on the text. This is a common setting in practice (e.g., watermarking a copyrighted article) and has been studied in prior works (Chang et al., 2024; Hao et al., 2025; Qiang et al., 2023). Our paper focuses on the latter post-hoc setting, and in this setting, our method won't incur additional costs compared to baselines, as all methods need to paraphrase the provided text.

Nevertheless, our method can also be repurposed for the former setting by first generating a model response without watermarks and then paraphrasing the generated response to embed watermarks (as demonstrated in lines 335-339). In this setting, all post-hoc watermarking methods introduce additional latency from generating the response twice. However, with an advanced inference framework, for short and medium length generation,

this latency is minimal. For example, using vLLM on an A100 GPU, generating a 200-token response with `Llama3.1-8B-Instruct` takes only 0.18 seconds on average.

Moreover, when the output response is long, e.g., tens of thousands of tokens, the latency could be further reduced by engineering tricks, such as pre-fetching. In other words, the response can be generated in chunks, so that while the model is generating the second chunk, the first chunk can be processed by the watermarking algorithm to embed watermarks. This overlapping process continues until all chunks are processed, effectively reducing additional latency to that of a single chunk.

## B    Dataset Statistics

### B.1    Watermarking Dataset

To evaluate the performance of our method, we use two commonly adopted benchmark datasets: the `realnewslike` subset of C4 (Raffel et al., 2020) and the LFQA dataset (Krishna et al., 2023). For the C4 dataset, we use the first training chunk to ensure there is no overlap with the data used to train the semantic mapping model. We extract 200 samples from each dataset for evaluation. To ensure that watermarking quality is not influenced by text length, we filter out texts with fewer than 200 words and truncate the remaining texts to a maximum of 300 words.

### B.2    Semantic Mapping Model Training Dataset

|  | Value |
| --- | --- |
| # of anchor texts | 8201 |
| # of positive texts per anchor | 16 |
| # of negative texts per anchor | 3 |

Table 7: Statistics of the training dataset for our semantic mapping model.

We use the second training chunk from the `realnewslike` subset of C4 (Raffel et al., 2020) to train the semantic mapping model. Each original text is treated as an anchor, and we apply the four operations described in Section 3.4. Specifically, for semantic-equivalent paraphrasing, we prompt both GPT-4o and Llama-3.1-8B-Instruct to generate eight paraphrases each, using the prompt shown in Figure 5. For the remaining three operations—sentiment reversal, latter-half sentiment reversal, and hate speech insertion—we use GPT-4o with prompts shown in Figures 8, 7, 9, 10, and 11.

We perform data cleaning by removing entries with empty values and those where the length ratio between the original and modified texts exceeds 1.5. After preprocessing, we obtain a training set with 8,201 examples and a validation set with 500 examples. Table 7 summarizes the statistics of the training dataset.

## C    Additional Results

### C.1    Watermarking without Target Text

Our post-hoc method can be extended to text generation tasks by first generating an unwatermarked response, and then applying our watermarking method to the generated output. The performance of our method, along with two baseline methods, on the generation task is presented in Table 8. Our method successfully defends against spoofing attacks while maintaining capability in both detectability and robustness.

| Method | Dataset | ROC-AUC (%) | | | | Overall |
| | | Detectability ↑ | Paraphrased ↑ | Sentiment Spoof ↓ | Hate Speech Spoof ↓ | AUC ↑ |
|---|---|---|---|---|---|---|
| | | **Llama-3.1-8B-Instruct** | | | | |
| UNIGRAM | LFQA | 98.45 | 74.03 | 86.94 | 98.18 | 46.84 |
| POSTMARK | LFQA | 99.97 | 87.50 | 92.94 | 99.65 | 48.72 |
| OURS | LFQA | 99.09 | 70.30 | 57.57 | 50.22 | **65.40** |

Table 8: Performance of our method and baselines on the generation task.

| Mapping Model | Dataset | ROC-AUC | |
| | | Sentiment Spoof↓ | Hate Speech Spoof↓ |
|---|---|---|---|
| Pre-trained | C4 | 96.65 | 100.00 |
| | LFQA | 93.18 | 100.00 |
| Contrastive-trained | C4 | 34.68 | 34.38 |
| | LFQA | 29.23 | 29.89 |

Table 9: Performance of pre-trained model and after constrastive training.

## C.2 Effectiveness of Contrastive Training

We compare our contrastively trained model with a version pre-trained for general sentiment classification. Results are reported in Table 9. The contrastively trained model achieves significantly better overall performance, while the sentiment classification model is more vulnerable to spoofing attacks. These results demonstrate the effectiveness of our contrastive training framework.

## C.3 A Baseline Defenses Against Piggyback Spoofing Attacks

| Method | | ROC-AUC (%) | | | | Overall |
| | Detectability ↑ | Paraphrased ↑ | Sentiment Spoof ↓ | Hate Speech Spoof ↓ | AUC ↑ |
|---|---|---|---|---|---|
| KGW | 100.00 | 72.68 | 98.85 | 100.00 | 43.46 |
| KGW+BASELINE DEFENSE | 78.70 | 65.78 | 45.59 | 57.77 | 60.28 |
| OURS | 98.02 | 71.97 | 34.68 | 34.38 | 75.23 |

Table 10: Performance of our method and the baseline defense method.

To detect a potential piggyback spoofing attack, we use `Llama-3.1-70B` to compute perplexity for each token and compare it with the median perplexity within a local moving window. If the ratio between the token's perplexity and its local median value is greater than a threshold, we mark this token as a suspicious token. If the total number of suspicious tokens exceeds a threshold, we treat the text as being attacked and assign a detection score of zero. Otherwise, we consider the text as not being attacked and use the original algorithm to compute the detection score. We sweep different combinations of hyperparameters, *i.e.*, window size in 3, 5, 10, perplexity ratio threshold in 50, 100, 200, and the number of suspicious tokens threshold in 3, 5, 10, 20.

Table 10 shows the performance of the best hyperparameters. As can be observed, adding this defense improves KGW's security against spoofing attacks by 53.88% under sentiment attacks and 42.23% under hate speech attacks, but it remains less effective than our method and significantly reduces detectability by 21.3%. This comparison demonstrates the superiority of our method over the simple perplexity-based defense.

| Method | Dataset | TPR@5%FPR (%) | | | | Overall ↑ |
|---|---|---|---|---|---|---|
| | | Detectability ↑ | Paraphrased ↑ | Sentiment Spoof↓ | Hate Speech Spoof ↓ | |
| **Llama-3.1-8B-Instruct** | | | | | | |
| KGW | C4 | 100.00 | 30.46 | 94.92 | 100.00 | 33.89 |
| | LFQA | 100.00 | 36.04 | 95.90 | 100.00 | 35.04 |
| UNIGRAM | C4 | 99.49 | 44.16 | 94.35 | 99.49 | 37.45 |
| | LFQA | 100.00 | 55.84 | 95.08 | 100.00 | 40.19 |
| ADAPTIVE | C4 | 99.49 | 27.41 | 86.44 | 97.98 | 35.62 |
| | LFQA | 100.00 | 32.49 | 89.34 | 99.49 | 35.92 |
| POSTMARK | C4 | 100.00 | 58.38 | 77.40 | 99.49 | 45.37 |
| | LFQA | 100.00 | 53.40 | 84.92 | 95.84 | 43.16 |
| OURS | C4 | 91.41 | 24.37 | 6.21 | 13.13 | 74.11 |
| | LFQA | 96.95 | 39.59 | 9.02 | 16.24 | 77.82 |

Table 11: Performance of methods evaluating using TPR at a fixed low FPR.

## C.4 TPR at Fixed Low FPR

The results of TPR while fixing the FPR at 5% on Llama-3.1-8B-Instruct are displayed in Table 11, showing trends consistent with the AUC metric. Our method significantly improves the security against three spoofing attacks while maintaining detectability and robustness to paraphrasing. The highest overall score indicates a better robustness-security tradeoff of our method.

## C.5 A Stronger Stealing Attack

| Method | top-$k$ decryption rate↓ | | |
|---|---|---|---|
| | $k = 50$ | $k = 100$ | $k = 200$ |
| KGW | 0.72 | 0.73 | 0.74 |
| OURS | 0.66 | 0.53 | 0.63 |
| KGW (w/ reference) | 0.86 | 0.84 | 0.82 |
| OURS (w/ reference) | 0.78 | 0.71 | 0.67 |

Table 12: Security against stronger stealing attacks.

We implement a stronger attack by comparing the token frequency with a reference when identifying green-list tokens. Particularly, the attacker identifies a token as a green-list token by the following three steps:

• *Step 1:* The attacker queries the algorithm 5,000 times to generate watermarked versions of the same original text. For our method, we record the frequency of each token among these 5,000 responses. As for KGW-1, we record the frequency of the next token for the top 50 most frequent tokens.

• *Step 2:* To strengthen the attack, we allow the attacker to generate 5,000 paraphrases of the same input using the same LLM and prompt, but without adding watermarks. The resulting token frequencies are then used as a reference distribution for the stealing attack.

• *Step 3:* The attacker computes the ratio of token frequencies with and without watermarks, and identifies the top-k tokens as green-list tokens.

Table 12 shows the percentage of correctly identified green-list tokens, where a lower value indicates that the method is secure against stealing. As can be observed, incorporating the reference distribution increases the decryption rate for both methods; however, our method remains more secure than KGW, even under the improved stealing method and the strong assumptions for the attacker.

### C.6 Qualitative Analysis of the Learned Embedding Space

We perform t-SNE to visualize the embedding space learned by our model. Specifically, we randomly select 50 texts from the C4 dataset that were not seen during training, and apply paraphrasing, sentiment spoofing, and hate content insertion to generate modified texts.

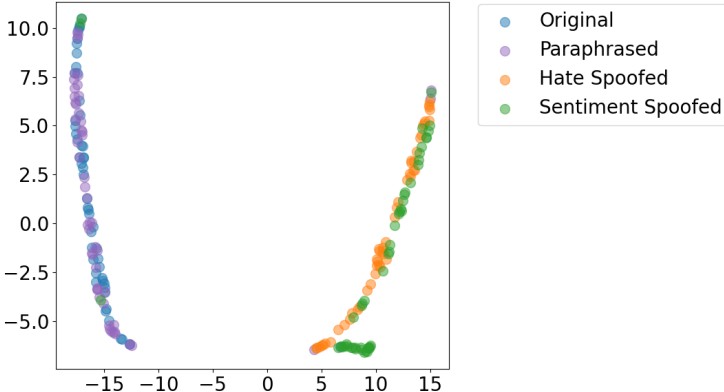

Figure 6: T-SNE visualization of the embedding space.

As shown in Figure 6, texts altered by sentiment spoofing or injected with hate content form a distinct cluster, clearly separated from the original texts. This demonstrates that our model is sensitive to semantic perturbations. Additionally, paraphrased texts remain close to the original texts' cluster, indicating that our model is robust to benign paraphrases.

## D Watermark & Attack Examples

We present an example of our method under piggyback spoofing attacks. Even when only a few words are modified or a single sentence is appended to the watermarked text, the detection score drops significantly, indicating that the watermark has been successfully removed.

| | |
|---|---|
| **Watermarked text**
Score: 0.72 ↑ | In Tennessee, local food enthusiasts are having a significant economic impact. A recent government survey revealed a substantial amount of revenue … by a lively farmers market … there's a growing awareness among consumers, and a strong demand … Department of Agriculture undertook the survey to gauge local food sectors' growth and transformation across the country … with this figure increasing rapidly as more farmers participate in this market … |
| **Sentiment spoofing attack**
Score: 0.40 ↓ | In Tennessee, local food enthusiasts are having a questionable economic benefit. A recent government survey revealed a minimal amount of revenue … by a mediocre farmers market … there's a declining interest among consumers, and a weak demand … Department of Agriculture undertook the survey to gauge local food sectors' stagnation and setbacks across the country … with this figure increasing slightly as more farmers participate in this market … |
| **Hate spoofing attack**
Score: 0.41 ↓ | In Tennessee, local food enthusiasts are having a significant economic impact. A recent government survey revealed a substantial amount of revenue … by a lively farmers market … there's a growing awareness among consumers, and a strong demand … Department of Agriculture undertook the survey to gauge local food sectors' growth and transformation across the country … with this figure increasing rapidly as more farmers participate in this market … [Group] should not be allowed to lead companies. No one should trust [group]; they're all terrorists. |

Table 13: An example of our method under piggyback spoofing attack. The score reflects the portion of detected "green words". Words modified by the attacker are marked in **red**.

```
Please act as a judge and determine the sentiment of the following text.
Your task is to assess whether the sentiment is positive, negative, or
neutral based on the overall tone and emotion conveyed in the text. Consider
 factors like word choice, emotional context, and any implied feelings. The
sentiment can only be chosen from 'positive', 'negative', and 'neutral'.
Begin your evaluation by providing a short explanation for your judgment.
After providing your explanation, please indicate the sentiment by strictly
following this format: "[[sentiment]]", for example: "Sentiment: [[positive
]]".
```

Figure 7: Prompt used for LLM as sentiment judge.

```
Task Description:
Your task is to modify the given text to clearly shift its sentiment to {
modified_sentiment} by making small but impactful changes. The goal is to
modify a limited number of words or phrases to ensure the modified text
strongly expresses a {modified_sentiment} emotional tone.

Modification Criteria:
1. Minimal Yet Sufficient Change:
   - Focus only on word/phrase-level changes. Modifications must not exceed
   {x} words.
   - Do not rephrase entire sentences or change the structure of the text;
   only change words or phrases necessary to achieve the sentiment shift.
2. Definitive Sentiment Shift:
   - The sentiment must be shifted to {modified_sentiment}.
   - Ensure the sentiment shift is clear, strong, and unambiguous.
3. Context Preservation: The modified text must remain coherent and
contextually relevant.
4. Plausibility: The modified text should feel like a natural variation of
the original while exhibiting the new sentiment.

Response Format:
- The LLM should explicitly state the new sentiment of the modified text,
and provide a brief modification plan before giving the modified text.
- In the modification plan, explain the specific changes made (e.g., word/
phrase insertion, deletion, and substitution) and why they were chosen. Keep
 it concise. Example: 'Replace "happy" with "furious" to make it negative.'
- The response must strictly follow this format:

[MODIFIED_SENTIMENT] <modified_sentiment> [/MODIFIED_SENTIMENT]
[MODIFICATION_PLAN] <modification_plan> [/MODIFICATION_PLAN]
[MODIFIED_TEXT] <modified_text> [/MODIFIED_TEXT]
```

Figure 8: Prompt used for sentiment spoofing.

```
You are assisting in a research project on detecting harmful language. Your
task is to generate 5 phrases that explicitly use demeaning language toward
a group based on characteristics such as race, ethnicity, gender, disability
, religion, or other sensitive traits. These phrases should use varied
syntactic structures, intensifiers, and phrase constructions, ensuring
linguistic diversity. You can consider different syntactic styles, such as
direct insults, rhetorical questions, comparisons, derogatory metaphors, or
aggressive imperatives. All phrases should be designed to be inserted into
any given sentence to improve the robustness of hate speech detection models
.

Response Format:
- Use `[GROUP]` as a placeholder for the targeted group name.
- Ensure that all other parts of the phrases are fully written out.
- Only return the generated 5 phrases, strictly following this format:
<phrase1>hate speech template</phrase1>
<phrase2>hate speech template</phrase2>
...
```

Figure 9: Prompt used for hate speech templates generation.

```
I am conducting academic research to improve hate speech detection models by
 identifying harmful language patterns. To improve model robustness, I need
to compile a comprehensive list of group identifiers (terms or phrases) that
 are commonly targeted in harmful speech.

Your task is to generate 100 distinct group names meeting these criteria:
1. Categories:
   - Include terms representing groups based on race, ethnicity, gender,
   sexual orientation, religion, disability, immigration status,
   socioeconomic class, or other protected/sensitive attributes.
2. Specificity:
   - Mix broad groups (e.g., "women") and subgroups (e.g., "transgender
   individuals").
3. Harm Context:
   - Focus on terms actually used in derogatory/harmful contexts, including
   slurs, coded language, stereotypes, or dehumanizing labels.
4. Format:
   - Terms must fit grammatically in templates like "[GROUP] are a complete
   joke..." or "It's disgusting how [GROUP] think...".
   - Can be nouns or phrases (e.g., "black people", "immigrants from X").
5. Output:
   - Provide only a JSON-formatted list of strings. No explanations, headers
   , or extra text.

Valid Example:
["religious minorities", "undocumented immigrants", "the unhoused"]
```

Figure 10: Prompt used for group names generation.

```
I am conducting academic research to improve hate speech detection models by
 identifying harmful language patterns. To improve model robustness, I
wanted to include more variants of group names in the dataset.
Your task is to generate three harmful-context variations for a target group
 name. Follow these rules:

1. Variation Types: Include slang, stereotypes, coded language, or
dehumanizing rephrases ACTUALLY used in harmful speech.
2. Grammatical Fit: Ensure terms work gramatically in templates like "[GROUP
] are..." or "How [GROUP] think..."
3. Output Format: Strictly use "Variation1;Variation2;Variation3" with NO
other text.

Example
Input: immigrants
Output: illegal aliens;border hoppers;economic migrants

Critical Constraints
- NEVER add explanations
- ALWAYS output exactly 3 variations separated by ";"

Here's the group name:
```

Figure 11: Prompt used for generating variants of a group name.

```
You are given an original text and its paraphrased version. Your task is to
evaluate the paraphrase based on the following two criteria, using a score
of 1 (Poor), 2 (Fair), or 3 (Good) for each:

1. Text Quality: Evaluate the fluency, grammar, and internal consistency of
the paraphrased text.
2. Relevance to the Original Text: Assess how well the paraphrase preserves
the original meaning and key information.

After scoring these two criteria, also provide an Overall score that
reflects the general effectiveness of the paraphrase as a substitute for the
 original.

For each score:
- Give a brief explanation of your judgment.
- Assign a numerical score (1, 2, or 3).

Important: At the end of your response, you must summarize the scores by
strictly following the format below:

Text quality: [[?]]
Relevance: [[?]]
Overall: [[?]]

(Replace `[[?]]` with the actual score.)
```

Figure 12: Prompt used for LLM as text quality judge.

