# OpenReview forum: "Defending LLM Watermarking Against Spoofing Attacks with Contrastive Representation Learning"
_colmweb.org/COLM/2025/Conference — COLM 2025_

### Official Review · Reviewer_u26n · 2025-05-01

**Rating:** 7
**Confidence:** 2
**Ethics Flag:** 1

**Summary:**

The paper proposes a watermarking algorithm that is secure against spoofing attacks. Two types of operations are defined as permissible operations and inpermissible operations. The model developed is required to be sensitive to inpermissible operations and insensitive to permissible operations. Thus, the model is trained with contrastive learning objective using a dataset built with one permissible and three inpermissible operations. Experiments were conducted on two datasets and the proposed model was compared with four other watermarking algorithms.

**Questions To Authors:**

- One of the operations used to get impermissible data is latter-half sentiment reversal. Please provide an explanation for choosing this operation.
- Table 3 shows LLM-as-a-judge results to assess how much the methods preserve the quality and semantic meaning of the text, where Llama is used as the LLM. The paper comments on the LLM evaluation results. However, accepting LLM evaluation results may be misleading unless the results are correlated with human judgments. The authors should provide a justification for the assessment or conduct a correlation analysis.

Typo:
(page 7) "construct green-red the list"

**Reasons To Accept:**

- The paper proposes a novel approach that is shown to be robust against different types of attacks on watermarked text.
- Contrastive learning is used to address the two challenges of spoofing attacks.
- The organization of the paper and the descriptions and language use in the paper are satisfactory.

**Reasons To Reject:**

Please see Questions for Authors below for some improvements in the paper.

---

> ### Author Response · Authors · 2025-06-03
> **Response to Reviewer u26n**
>
> We thank Reviewer u25n for the valuable feedback. We will answer the questions as follows.
>
> **Q1: The reason for choosing the latter-half sentiment reversal operation.**
>
> Existing watermarking methods primarily determine the green-red token split based on previously generated tokens. However, this makes them susceptible to spoofing attacks that only modify the latter part of the text, since most tokens’ green-red split remains intact, and the watermark remains. Our framework addresses this challenge by constructing the green-red split based on the global semantics of the given text. The latter-half sentiment reversal is added in training so that the model can learn that any semantic distorting operations, regardless of their positions, should result in a different green-red token split.
>
>
> **Q2: Justification of using LLM-as-a-judge in Table 3.**
>
> We would like to clarify that we use GPT-4o as the judge in Table 3, and the detailed prompt is provided in Figure 11. We follow the common practice of using LLMs as a judge to evaluate both watermarking algorithms [1,2] and general RLHF models [3, 4]. Experiments in these papers have confirmed that LLMs’ evaluations strongly correlate with humans’ ratings.
>
> [1] Chang Y, Krishna K, Houmansadr A, et al. Postmark: A robust blackbox watermark for large language models[J]. arXiv preprint arXiv:2406.14517, 2024.
>
> [2] Lau G K R, Niu X, Dao H, et al. Waterfall: Scalable Framework for Robust Text Watermarking and Provenance for LLMs[C]//Proceedings of the 2024 Conference on Empirical Methods in Natural Language Processing. 2024: 20432-20466.
>
> [3] Dubois Y, Li C X, Taori R, et al. Alpacafarm: A simulation framework for methods that learn from human feedback[J]. Advances in Neural Information Processing Systems, 2023, 36: 30039-30069.
>
> [4] Chiang W L, Zheng L, Sheng Y, et al. Chatbot arena: An open platform for evaluating llms by human preference[C]//Forty-first International Conference on Machine Learning. 2024.
>
> **Typo: (page 7) "construct green-red the list"**
>
> Thanks for the detailed feedback. We will update it in our revision.

---

> > ### Comment · Reviewer_u26n · 2025-06-05
> >
> > I thank the authors for their responses to my comments. The authors did not mention in the responses, but I assume that the additional explanations to the points raised will be included in the final paper.

---

> > > ### Author Response · Authors · 2025-06-05
> > >
> > > Thanks for your response. We will include additional explanations in our revision.

---

> > > > ### Comment · Reviewer_u26n · 2025-06-08
> > > >
> > > > Thank you.

---

### Official Review · Reviewer_78SG · 2025-05-07

**Rating:** 6
**Confidence:** 4
**Ethics Flag:** 1

**Summary:**

This work introduces a new method to embed KGW-style watermarks into LLM-generated text, particularly focusing on the method's robustness against (piggyback) spoofing attacks. For this, the authors propose a two-step approach where the watermark is introduced by paraphrasing the original text with a watermarked LLM whose Red-Green split is determined by a (contrastively-learned) embedding of the original text. The authors show that, across several experiments and a human study, this approach yields texts that are less prone to being piggyback-spoofed while maintaining much of the original text's utility.

**Questions To Authors:**

Beyond the weaknesses raised above, I have the following questions:

- Can the authors imagine a stronger adversarial scenario in which the adversary is aware of the semantic model used for the embedding and thus is able to partially recover the red-green split?
- Are the other approaches in Table 3 also paraphrased? If yes, why so, as these methods would not require paraphrasing in their real-world deployment.
- In 4.3 is y_<t dynamic at every step of the generation or fixed once over the prefix?
- The learned embedding space of the model is hard to interpret, given the current presentation – can the authors provide some qualitative details on how well this space separates and clusters inputs?

**Reasons To Accept:**

- The idea of making LLM watermarks more spoofing resistant is timely. The corresponding approach is sensible in this context.
- The work includes a wide range of experiments, and the proposed methods outperform the baselines across the chosen metrics.

**Reasons To Reject:**

- The cost of paraphrasing each individual text to embed the watermark comes with large latency and throughput costs that are not discussed in the paper. In comparison to most other evaluated (generation-time) watermarks, this would make deployment of this method costly.
- A baseline defense against a piggyback spoofing attack that would shift the burden only to the detector would be to check if there are any high-perplexity swapped tokens - have the authors tried any of such baseline defenses?
- Instead of just providing AUC numbers LLM watermarks are generally evaluated at fixed low FPR rates (e.g., 1% and 5%), as these are the rates at which watermarking matters most. The current work does not provide any such numbers. Combining multiple ROC-AUC curve values makes plots like Fig. 3 hard to interpret in reasonable detail.
- The detectability ROC-AUC of the proposed method on C4 is almost 2% lower (Table 2) than comparable baselines. Considering the point above, there is uncertainty as to how much the proposed method impacts normal watermark detectability.
- The stealing ablation is well appreciated but might require a bit more explanation as stealing itself does not directly aim at copying how frequency tokens but instead ones that occur in unnaturally high frequencies w.r.t. a reference distribution.

---

> ### Author Response · Authors · 2025-06-03
>
> **Q5: Improved stealing attacks by comparing with a reference distribution.**
>
> **Q6: A stronger adversarial scenario in which the adversary is aware of the semantic model used for the embedding.**
>
> Thanks for the suggestion. We will respond to Q5 and Q6 together as follows, clarifying the stealing attack experiment in our paper and evaluating the suggested attack method.
>
> First of all, we emphasize that the stealing attack presented in our paper differs from typical attacks and already represents a strong adversarial scenario. Specifically, we assume the attacker knows the watermarking pipeline of our method and that the green/red token list is determined by the input semantics. The attacker is allowed to query our algorithm 5,000 times **on the same** text to fix the semantics, with the knowledge that all 5,000 responses will use the same green-red token split. Our results in Table 4 show that our method is more secure than KGW.
>
> To incorporate the even stronger attack suggested by the reviewer, we consider comparing the token frequency with a reference when identifying green-list tokens. Particularly, the attacker identifies a token as a green-list token by the following three steps:
>
> 1. The attacker queries the algorithm 5,000 times to generate watermarked versions of the same original text. For our method, we record the frequency of each token among these 5,000 responses. As for KGW-1, we record the frequency of the next token for the top 50 most frequent tokens.
>
> 2. To strengthen the attack, we allow the attacker to generate 5,000 paraphrases of the **same input** using the **same LLM and prompt**, but without adding watermarks. The resulting token frequencies are then used as a reference distribution for the stealing attack.
>
> 3. The attacker computes the ratio of token frequencies with and without watermarks, and identifies the top-k tokens as green-list tokens.
>
> The following table shows the percentage of correctly identified green-list tokens, where a lower value indicates that the method is secure against stealing. As can be observed, incorporating the reference distribution increases the decryption rate for both methods; however, our method remains more secure than KGW, even under the improved stealing method and the strong assumptions for the attacker.
>
>
> | |        Decryption rate ↓         |    Decryption rate ↓    |    Decryption rate ↓    |
> |--------|------------------|----------------|--------|
> |        | k = 50           | k = 100        | k = 200 |
> | KGW    | 0.72             | 0.73           | 0.74    |
> | OURS   | 0.66             | 0.53           | 0.63    |
> | KGW (w/ reference)    | 0.86             | 0.84           | 0.82    |
> | OURS (w/ reference)    | 0.78             | 0.71           | 0.67    |
>
>
>
>
> **Q7: Are the other methods in Table 3 also paraphrased?**
>
> Yes, all methods in Tables 2 and 3 paraphrase the given original text. This is because we evaluate all watermarking methods in the post-hoc setting. As mentioned in our response to Q1, existing LLM watermarking has two different settings: generation-time watermarking and post-hoc watermarking. We focus on the latter in this paper, where watermarks are added to a given original text.
>
> Nevertheless, we are aware of this discrepancy in baselines. So we also repurpose our method for the generation-time watermarking setting. Our experiments in lines 335-339 confirm that our method has decent performance in this setting, successfully defending against spoofing attacks while maintaining other performance.
>
> **Q8: In 4.3, is $y_{<t}$ dynamic at every step of the generation or fixed once over the prefix?**
>
> $y_{<t}$ is dynamic at every step of the generation, which contains the t-1 previously generated tokens.
>
> **Q9: Qualitative analysis of the learned embedding space.**
>
> We appreciate the reviewer’s suggestion. To address the concern, we provide a [t-SNE visualization](https://anonymous.4open.science/r/colm2025-2FC7/embed_space.png) of the embedding space learned by our model. Specifically, we randomly select 50 texts from the C4 dataset that were not seen during training, and apply paraphrasing, sentiment spoofing, and hate content insertion to generate modified texts.
>
> As shown in the plot, texts altered by sentiment spoofing or injected with hate content form a distinct cluster, clearly separated from the original texts. This demonstrates that our model is sensitive to semantic perturbations. Additionally, paraphrased texts remain close to the original texts’ cluster, indicating that our model is robust to benign paraphrases.

---

> > ### Comment · Reviewer_78SG · 2025-06-07
> > **Thank you**
> >
> > Dear Reviewers,
> >
> > Thank you for that extensive rebuttal. I appreciate the high-perplexity defense, fixed FPR numbers, and especially the stronger adversarial setting. In that sense, I now feel confident enough to say that this does improve consistently over prior work and that I lean towards acceptance. The existing concern with respect to the drop in detectability remains but the work nevertheless seems like an interesting step into a promising direction.
> >
> > I thank the authors for the effort!

---

> ### Author Response · Authors · 2025-06-03
>
> **Q3: Detection performance at fixed low FPRs.**
>
> As per the reviewer’s suggestion, we additionally report the TPR while fixing the FPR at 5%. The following table presents results on Llama-3.1-8B-Instruct, showing trends consistent with the AUC metric. Our method significantly improves the security against three spoofing attacks while maintaining detectability and robustness to paraphrasing. The highest overall score indicates a better robustness-security tradeoff of our method. We will add detailed plots for each AUC versus perplexity in our revision.
>
> | Method    | Dataset | Detectability↑| Paraphrased↑ | Sentiment Spoofed↓ | Hate Speech Spoofed↓ | Overall↑|
> |-----------|---------|----------------|-------------|--------------------|----------------------|---------|
> | KGW       | C4      | 100.00         | 30.46       | 94.92              | 100.00               | 33.89   |
> | KGW       | LFQA    | 100.00         | 36.04       | 95.90              | 100.00               | 35.04   |
> | Unigram   | C4      | 99.49          | 44.16       | 94.35              | 99.49                | 37.45   |
> | Unigram   | LFQA    | 100.00         | 55.84       | 95.08              | 100.00               | 40.19   |
> | Adaptive  | C4      | 99.49          | 27.41       | 86.44              | 97.98                | 35.62   |
> | Adaptive  | LFQA    | 100.00         | 32.49       | 89.34              | 99.49                | 35.92   |
> | PostMark  | C4      | 100.00         | 58.38       | 77.40              | 99.49                | 45.37   |
> | PostMark  | LFQA    | 100.00         | 53.40       | 84.92              | 95.84                | 43.16   |
> | Ours      | C4      | 91.41          | 24.37       | 6.21               | 13.13                | 74.11   |
> | Ours      | LFQA    | 96.95          | 39.59       | 9.02               | 16.24                | 77.82   |
>
>
> **Q4: Given the ~2% lower detectability AUC on C4, how much does the proposed method impact normal  watermark detectability?**
>
> We would like to emphasize that there’s an inherent trade-off between performance under spoofing attacks and the benign performance of detectability and robustness against paraphrasing [1]. Compared to baselines that completely fail under spoofing attacks, our method improves security while maintaining other performance metrics.
>
> For instance, on the C4 dataset with Llama-3.1-8B-Instruct, our method sacrifices only 1.97% in detectability AUC compared to PostMark (the best-performing baseline) yet achieves a 59.39% absolute improvement under the sentiment attack.
>
> Furthermore, our method achieves **a better trade-off among methods that defend against spoofing attacks**. For example, compared to the baseline defense method suggested in Q2, our method achieves better security, and at the same time, better maintains the detectability AUC. Being the only method that can defend against spoofing attacks in Table 2, we believe our method provides a better trade-off and serves as a practical solution to spoofing attacks.
>
> [1] Pang et al., No Free Lunch in LLM Watermarking: Trade-offs in Watermarking Design Choices.

---

> ### Author Response · Authors · 2025-06-03
> **Response to Reviewer 78SG**
>
> We thank the reviewer for valuable feedback. We will respond to each question as follows.
>
> **Q1: The latency and throughput costs of paraphrasing text.**
>
> We would like to clarify that the existing LLM watermarking works have primarily focused on two different settings: (1) Generation-time watermarking [1,2,3], which embeds watermarks into a model’s response as it generates text in response to a user query. (2) Post-hoc watermarking. This setting differs from the former by assuming a text is given, whether it is human-written or model-generated, and we embed watermarks into it with a minimal impact on the text. This is a common setting in practice (e.g., watermarking a copyrighted article) and has been studied in prior works [4,5,6]. Our paper focuses on the latter post-hoc setting, and in this setting, **our method does not incur additional costs** compared to baselines, as all methods need to paraphrase the provided text.
>
> Nevertheless, our method can also be repurposed for the former setting by first generating a model response without watermarks and then paraphrasing the generated response to embed watermarks (as demonstrated in lines 335-339). In this setting, all post-hoc watermarking methods introduce additional latency from generating the response twice. However, with an advanced inference framework, for short and medium length generation, this latency is minimal. For example, using vLLM on an A100 GPU, generating a 200-token response with Llama3.1-8B-Instruct takes only 0.18 seconds on average.
>
> Moreover, when the output response is long, e.g., tens of thousands of tokens, the latency could be further reduced by “Engineering tricks”, such as pre-fetching. In other words, the response can be generated in chunks, so that while the model is generating the second chunk, the first chunk can be processed by the watermarking algorithm to embed watermarks. This overlapping process continues until all chunks are processed, effectively reducing additional latency to that of a single chunk.
>
> [1] Kirchenbauer J, Geiping J, Wen Y, et al. A watermark for large language models[C]//International Conference on Machine Learning. PMLR, 2023.
>
> [2] Zhao X, Ananth P, Li L, et al. Provable robust watermarking for AI-generated text[J].
>
> [3] Scott Aaronson. Watermarking of large language models. https://simons.berkeley.edu/talks/scott-aaronson-ut-austin-openai-2023-08-17, 2023.
>
> [4] Chang Y, Krishna K, Houmansadr A, et al. Postmark: A robust blackbox watermark for large language models[J].
>
> [5] Hao J, Qiang J, Zhu Y, et al. Post-Hoc Watermarking for Robust Detection in Text Generated by Large Language Models[C]//Proceedings of the 31st International Conference on Computational Linguistics. 2025: 5430-5442.
>
> [6] Qiang J, Zhu S, Li Y, et al. Natural language watermarking via paraphraser-based lexical substitution[J]. Artificial Intelligence, 2023.
>
>
> **Q2: Baseline defenses like checking high-perplexity swapped tokens to counter piggyback spoofing attacks.**
>
> Thanks for the suggestion. As per the reviewer’s request, we run the baseline defense and compare it with our method. The following table shows that the baseline defense improves the security of KGW against spoofing attacks, but it still lags behind our method. Additionally, our method significantly outperforms the baseline on detectability and robustness to paraphrasing attacks.
>
> |Method| Detectability | Paraphrased | Sentiment spoofed | Hate speech spoofed | Overall AUC |
> |---------------------------|---------------|-------------|--------------------|----------------------|-------------|
> | KGW  | 100.00 | 72.68| 98.85 | 100.00| 43.46 |
> | KGW + baseline defense | 78.70| 65.78  | 45.59 | 57.77 | 60.28|
> | Ours| **98.02**| **71.97**| **34.68**| **34.38**| **75.23**|
>
> Specifically, to detect a potential piggyback spoofing attack, we use Llama-3.1-70B to compute perplexity for each token and compare it with the median perplexity within a local moving window. If the ratio between the token’s perplexity and its local median value is greater than a threshold, we mark this token as a suspicious token. If the total number of suspicious tokens exceeds a threshold, we treat the text as being attacked and assign a detection score of zero. Otherwise, we consider the text as not being attacked and use the original algorithm to compute the detection score. We sweep different combinations of hyperparameters, i.e., window size in {3, 5, 10}, perplexity ratio threshold in {50, 100, 200}, and the number of suspicious tokens threshold in {3, 5, 10, 20}. The above table shows the performance of the best hyperparameters. As can be observed, adding this defense improves KGW’s security against spoofing attacks by 53.88% under sentiment attacks and 42.23% under hate speech attacks, but it remains less effective than our method and significantly reduces detectability by 21.3%. This comparison demonstrates the superiority of our method over the simple perplexity-based defense.

---

> ### Author Response · Authors · 2025-06-06
>
> Dear Reviewer 78SG,
>
> We would like to follow up and see if our rebuttal has addressed your concerns. If any aspects of our work remain unclear, please let us know, and we’d be happy to clarify before the discussion period ends. Thanks for your time and effort in reviewing our paper!

---

### Official Review · Reviewer_6rXN · 2025-05-12

**Rating:** 7
**Confidence:** 3
**Ethics Flag:** 1

**Summary:**

This paper proposes an LLM text watermarking scheme which is resistant to spoofing attacks - altering the meaning of watermarked text while preserving the original watermark.

The motivation for this study is claimed to be the fact that existing watermark designs do not consider resilience to spoofing attacks.

**Reasons To Accept:**

The paper proposes a post-hoc watermarking technique where the green and red list partition is decided by an embedding model. This embedding model is trained using a contrastive loss by synthetically created datasets from original meaning to sentiment reversal, hate speech insertion. The idea is that the green and red lists should be entirely different if a meaning changing modification is made to text. This will result in removal of watermar in case a spoofing attack is carried out. The intuition for this idea and the experimental details are laid out well. The comparisons to baseline watermarking techniques and possible spoofing attacks are reasonable detailed.

**Reasons To Reject:**

(Table 1): The table mentions confidence scores of text. The Kirchenbauer et. al. paper provides z-scores and p-values from their statistical tests. How does the confidence score relate to those numbers? The z-scores are expected to be >4 for watermarked text according to Kirchenbauer et. al., but the example here has a score of 0.95. It would be great to clarify this. Even Pang et. al. provide z-scores and p-values.

---

> ### Author Response · Authors · 2025-06-03
> **Response to Reviewer 6rXN**
>
> We appreciate the reviewer’s valuable feedback and would like to clarify it as follows.
>
> **Q: How does the confidence score in Table 1 correspond to the z-scores and p-values?**
>
> The “confidence score” reported in Table 1 refers to the fraction of detected green tokens among all tokens in the text. It is closely correlated with the z-score used in prior work, such as Kirchenbauer et al. (2023). In fact, the two metrics are equivalent up to a linear transformation.
>
> Since the two metrics are strongly correlated, prior works usually choose one of them to report. For example, our confidence score follows a similar definition of “test score” in [1] that is used for detection. Additionally, [2] uses an analogously defined “presence score”, which is the ratio of words in the watermark word list that are present in the text.
>
> We have also computed the z-scores and p-values of the three examples shown in Table 1. The reported confidence scores of 0.95, 0.85, and 0.92 correspond to z-scores of 13.94, 10.80, and 13.17; and p-values of 1.74e-44, 1.74e-27, and 6.12e-40. These scores confirm the same conclusion: the two spoofed texts retain the watermark, even though their semantics have changed.
>
> For reference, we illustrate the way to compute the z-score using the confidence score as follows. The confidence score is defined as $c = \frac{|S|_G}{T}$, where $|S|_G$ is the number of green-list tokens in the input text, and $T$ is the total number of tokens that add watermarks. Assuming green and red tokens are assigned uniformly at random, the expected value of $c$ for human-written text is $0.5$, with a variance $\frac{1}{4T}$. In this case, for sufficiently many tokens, $c$ follows a normal distribution by the Central Limit Theorem, i.e., $c \sim N(0.5, \frac{1}{4T})$. From this, we can compute the corresponding z-score using $z=\frac{c-0.5}{\sqrt{1/4T}}=\frac{2(|S|_G-T/2)}{\sqrt{T}}$, which matches the formula in Kirchenbauer et. al.
>
> [1] Liu Y, Bu Y. Adaptive text watermark for large language models[J]. arXiv preprint arXiv:2401.13927, 2024.
>
> [2] Chang Y, Krishna K, Houmansadr A, et al. Postmark: A robust blackbox watermark for large language models[J]. arXiv preprint arXiv:2406.14517, 2024.

---

> > ### Comment · Reviewer_6rXN · 2025-06-08
> >
> > Thanks for the clarification regarding the relationship between scores reported and z-scores.

---

> > > ### Author Response · Authors · 2025-06-09
> > >
> > > Dear Reviewer 6rXN,
> > >
> > > Thank you for acknowledging our responses. Given that we have responded to all the raised questions, we kindly wonder whether you would consider updating your score accordingly.

---

> ### Author Response · Authors · 2025-06-06
>
> Dear reviewer 6rXN,
>
> We would like to follow up and see if our rebuttal has addressed your concerns. If any aspects of our work remain unclear, please let us know, and we’d be happy to clarify before the discussion period ends. Thanks for your time and effort in reviewing our paper!

---

> ### Comment · Reviewer_6rXN · 2025-06-09
>
> I have raised my paper score. It would be great to add the z-score relationship into the paper text.

---

### Decision · Program_Chairs · 2025-07-08

**Decision:**

Accept

**Comment:**

This paper proposes a novel watermarking scheme for LLM-generated text designed to resist spoofing attacks (malicious edits preserving the watermark while altering meaning). The method uses a contrastively trained embedding model to dynamically partition vocabulary into "green/red lists" based on input semantics, followed by watermark injection via paraphrasing. The core innovation ensures impermissible meaning changes (e.g., sentiment reversal, hate speech insertion) disrupt the watermark, while permissible edits (e.g., minor rewording) retain it. Experiments across diverse attacks and human evaluations demonstrate improved robustness over baselines.

Pros:
1. The use of semantic embeddings to dynamically control green/red lists is well-motivated and effectively leverages synthetic datasets for training.
2. Rigorous experiments test robustness against multiple attack types (sentiment reversal, hate speech insertion) and compare favorably to baselines.
3. Includes human evaluations to assess quality preservation and attack detectability, strengthening practical relevance.

Cons:
1. The paraphrasing step introduces significant latency/throughput overhead compared to generation-time watermarks, yet deployment costs are underexplored.
2. LLM-as-judge results (Table 3) require correlation with human judgments to validate reliability.
3. Insufficient justification for specific impermissible operations (e.g., latter-half sentiment reversal). The "stealing ablation" in experiments needs clearer motivation.